# Liver Kinase B1—A Potential Therapeutic Target in Hormone-Sensitive Breast Cancer in Older Women

**DOI:** 10.3390/cancers11020149

**Published:** 2019-01-28

**Authors:** Binafsha Manzoor Syed, Andrew R Green, David A L Morgan, Ian O Ellis, Kwok-Leung Cheung

**Affiliations:** 1Nottingham Breast Cancer Research Centre, School of Medicine, University of Nottingham, Nottingham DE22 3DT, UK; drbinafsha@hotmail.com (B.M.S.); andrew.green@nottingham.ac.uk (A.R.G.); ian.ellis@nottingham.ac.uk (I.O.E.); 2Medical Research Centre, Liaquat University of Medical & Health Sciences, Jamshoro 71000, Pakistan; 3Department of Oncology, Nottingham University Hospitals, Nottingham NG5 1PB, UK; dalmorgan@me.com

**Keywords:** LKB1, Breast Cancer, Older women, Metformin, Endocrine therapy

## Abstract

*Background*: The role of liver kinase B1 (LKB1), a serine/threonine kinase, has been described in the development of PeutzJagher’s syndrome, where a proportion (~45%) of patients have developed breast cancer in their lifetime. Cell line studies have linked LKB1 with oestrogen receptors (ER) and with the Adenosine monophosphate-activated protein kinase (AMPK) pathway for energy metabolism. However, limited studies have investigated protein expression of LKB1 in tumour tissues and its intracellular relationships. This study aimed to investigate the intracellular molecular relationships of LKB1 in older women with early operable primary breast cancer and its correlation with long-term clinical outcome. *Methods*: Between 1973 and 2010, a consecutive series of 1758 older (≥70 years) women with T0-2N0-1M0 breast carcinoma were managed in a dedicated facility. Of these, 813 patients underwent primary surgery, and 575 had good quality tumour samples available for tissue microarray construction. LKB1 was assessed in 407 cases by indirect immunohistochemistry (IHC). Tumours with 30% or more of cells with cytoplasmic LKB1 expression were considered positive. LKB1 expression was compared with tumour size, histological grade, axillary lymph node stage, ER, PgR, EGFR, HER2, HER3, HER4, BRCA1&2, p53, Ki67, Bcl2, Muc1, E-Cadherin, CD44, basal (CK5, CK5/6, CK14 and CK17) and luminal (CK7/8, CK18 and CK19) cytokeratins, MDM2 and MDM4, and correlated with long-term clinical outcome. *Results*: Positive LKB1 expression was seen in 318 (78.1%) patients, and was significantly associated with high tumour grade, high Ki67, over-expression of HER2, VEGF, HER4, BRCA2, MDM2 and negative expression of CD44 (p < 0.05). There was no significant correlation with tumour size, axillary lymph node status, ER, PgR, p53, basal or luminal cytokeratins, Bcl2, Muc1, EGFR, HER3, MDM4, E-cadherin and BRCA1. LKB1 did not show any significant influence on survival in the overall population; however, in those patients receiving adjuvant endocrine therapy for ER positive tumours, those with positive LKB1 had significantly better 5-year breast cancer specific survival when compared to those without such expression (93% versus 74%, *p* = 0.03). *Conclusion*: LKB1 expression has shown association with poor prognostic factors in older women with breast cancer. However, LKB1 expression appears to be associated with better survival outcome among those patients receiving adjuvant endocrine therapy. Further research is required to explore its potential role as a therapeutic target.

## 1. Introduction

Liver kinase B1 (LKB1) or serine/threonine kinase-11, a 436-amino acid chain protein with a molecular weight of 49 kDa, is associated with the development of Peutz-Jeghers syndrome (PJS) [1,2]. In addition, approximately 45% of PJS patients have been reported to develop breast cancer in their lifetime [3]. LKB1 is known to work as a tumour suppressor gene as well as an upstream kinase which is linked to the cellular functions during normal state and in metabolic stress response [4,5]. LKB1 is normally located in the nucleus, however, upon activation, it is relocated to the cytoplasm, where it acts as a co-activator of the oestrogen receptor (ER), controls the citric acid cycle (KREB’s cycle) and cell energy metabolism via the adenosine mono-phosphate activated protein kinase (AMPK) pathway, and inhibits the production of the aromatase enzyme [6,7,8,9,10,11]. It has also been reported to control the cell cycle at the G0–G1 check point via the p53 pathway, to be associated with angiogenesis through the control of Vascular Endothelial Growth Factor (VEGF) and with cell polarity [8,11,12]. A proposed mechanism of LKB1 function in breast cancer cells is presented in Figure 1. Previously, reported data linked LKB1 with poor prognostic factors [11]. A cell line study on MDA-MB-231 cells showed an association of LKB1 with gemcitabine resistance [13]. Additionally, cell line studies suggested that LKB1 is involved in DNA repair, thus deficient cells demonstrate delayed repair and respond well to DNA-based therapy, such as PARP inhibitors [14]. Cell-based studies showed that LKB1-null cells possess invasion and breast cancer stem cell like properties. However, LKB1 enhancing therapy (i.e., Honokoil, a bioactive molecule) showed improving outcome by reducing cellular invasiveness and stemness [15], thus suggesting a therapeutic significance. 

Currently available literature is based on in vitro or in vivo model studies, or gene technologies, such as Western blot. However, there is limited information available on the subject of using immunohistochemistry (IHC), which is currently a widely used technique for analysis of therapeutic targets in clinical practice. This study was designed to analyse the association of LKB1 with other biological markers of known significance in breast cancer and to correlate its expression with long-term clinical outcome in older women with primary breast cancer.

## 2. Materials and Methods

### 2.1. Patients

Over 37 years (1973–2010), 1758 older (≥70 years) women with early operable primary breast cancer (T0-2, N0-1, M0) were managed in a dedicated facility in a single institution with clinical information available from diagnosis till death/last follow-up. Eight hundred and thirteen patients underwent primary surgery (with optimal adjuvant therapy as per unit policy at the time [17]) and among them, good quality formalin-fixed paraffin-embedded surgical specimens were available from 575 patients for tissue microarray (TMA) construction. The management pattern over this period of time was evolving, a detailed description of which was reported earlier [17]. Briefly during the 1970s and 1980s, ER status was not available and there was no consensus on recommendations for adjuvant systemic therapy. During the 1990s, adjuvant endocrine therapy was advised based on the clinical judgement of the treating physician and/or the patient’s choice. In the recent decade, there have been structured recommendations for the use of adjuvant endocrine therapy. Those patients with ER positive tumours and Nottingham Prognostic Index (NPI) <3 (excellent prognostic group) were not given any adjuvant systemic therapy, those with an NPI of 3–3.4 (good prognostic group) were offered endocrine therapy and given the choice of tamoxifen or a non-steroidal aromatase inhibitor, and those with an NPI of >3.4 (moderate to poor prognostic group) were given a non-steroidal aromatase inhibitor. Adjuvant chemotherapy was considered in fit, relatively younger patients with moderate to poor prognostic tumours, in particular those with ER negative tumours. Trastuzumab was considered in relatively fit patients with HER2 positive tumours having moderate to poor prognosis. 

### 2.2. Tissue Microarray Construction 

Tissue microarrays (TMAs) of formalin-fixed paraffin-embedded tumour sections were constructed as described [18]. Briefly, 0.6 mm-diameter cores of the representative part of the tumour blocks were implanted in the TMA blocks using Beecher’s manual tissue microarrayer (MP06 Beecher Instruments Inc, Sun Prairie, WI, USA). 

### 2.3. Immunohistochemistry

LKB1 and 24 other biological markers including ER, progesterone receptor (PgR), Epidermal Growth Factor Receptor (EGFR), Human Epidermal Growth Factor Receptor (HER)-2, HER3, HER4, Breast Cancer Associated gene (BRCA)1&2, p53, Ki67, B-Cell Lymphoma (Bcl)2, Mucin (MUC)1, E-Cadherin, basal (CK5, CK5/6, CK14 and CK17) and luminal (CK7/8, CK18 and CK19) cytokeratins, and Mouse Double Minute (MDM)2 and MDM4 were analysed using indirect IHC by StreptAvidin Biotin Complex and EnVision methods as described [19]. 

### 2.4. Scoring 

Immunohistochemical staining of biological markers was assessed by the percentage of invasive tumour cells stained and by McCarty’s immunohistochemical scoring (H-score) (range 0–300) [20]. The LKB1 expression of 30% or more positive cells was considered positive (Figure 2a–d). Previously, studies have shown that the active form of LKB1 is found in the cytoplasm and is associated with other markers and outcome of breast cancer; thus, cytoplasmic expression of LKB1 was considered in our study [21]. 

### 2.5. Statistical Analysis

X-tile bio-informatics software (Yale University, New Haven, CT, USA) was used to define cut-offs for positivity of the expression of the biological markers based on breast cancer specific survival (BCSS) [22]. The cut-offs for all the biological markers were determined by using the same metric in X-tile. The cut-off values to define positive expression are the same as reported earlier [23]. The Statistical Package for Social Sciences (SPSS, version 16.0, Chicago, IL, USA) was used for data collection and analysis. A Chi-squared test was used for the analysis of the association of the LKB1 with other biomarkers in terms of expression. The clinical outcome was evaluated in terms of BCSS, metastases free survival, local recurrence free survival and regional recurrence free survival, breast cancer specific survival was defined as survival from the date of diagnosis till death from breast cancer; metastases free, local recurrence free and regional recurrence free survivals were calculated from the date of diagnosis till the appearance of the respective recurrences. Survival was analysed by using Kaplan-Meier methods with application of log-rank and generalised Wilcoxon tests, as appropriate. A *p*-value of <0.05 was considered significant. 

### 2.6. Ethical Consideration

The study was approved by the Nottingham Research Ethics Committee 2 under ethical approval number C2020313. 

## 3. Results

### 3.1. Patients

Of the 575 patients who had TMAs constructed from surgical specimens, 168 cores were lost during the IHC process and 407 cases had LKB1 measured. The median age of the patients was 76 (range 70–91) years, with the majority having had no axillary lymph node involvement (54.2%). Most of the patients (*n* = 296 (72.7%)) underwent a mastectomy and 46.6% of patients (*n* = 189) received adjuvant endocrine therapy. None of these patients received chemotherapy. The median follow-up was 60 months (longest = 261 months). 

### 3.2. Pattern of LKB1 Expression and Association with Other Clinico-Pathological Parameters and Biological Markers

Overall, 78.1% (*n* = 318) of patients showed positive cytoplasmic expression of LKB1 in their tumours. 

#### 3.2.1. Pathological Parameters

The expression of LKB1 was significantly associated with high histological grade (among LKB1 positive, grade 1&2 = 44.5%, grade 3 = 55.5%; among LKB1 negative, grade 1&2 = 59.2% and grade 3 = 40.8%, *p* = 0.01).However there was no statistically significant correlation with tumour size(among LKB1 positive, <3 cm = 81.7% and among LKB1 negative, <3 cm = 75.6%, *p* = 0.15) and axillary lymph node status (among LKB1 positive:, stage 1&2 = 87.1%, stage 3 = 12.9%, among LKB1 negative, stage 1&2 = 89.5%, stage 3 = 10.5%, *p* = 0.41).

#### 3.2.2. Biological Markers

The positive expression of LKB1 was significantly associated with positive expression of HER2 (*p* = 0.003), Ki67 (*p* = 0.01), VEGF (*p* = 0.002), HER4 (*p* = 0.001), BRCA2 (*p* = 0.01) and MDM2 (*p* < 0.001), and negative expression of CD44 (*p* = 0.03). However, there was no statistically significant correlation with, ER, PgR, p53, basal or luminal cytokeratins, Bcl2, Muc1, EGFR, HER3, MDM4, E-cadherin, and BRCA1.A, as summarised in Table 1.

### 3.3. Correlation with Molecular Classes of Breast Cancer

The expression of LKB1 was correlated with molecular classes (subtypes) of breast cancer found in older women (published previously [23]). The results showed no significant difference in the expression across them (Table 1).

### 3.4. Association of LKB1 with Clinical Outcome

#### Breast Cancer Specific Survival (BCSS)

At a median follow-up of 60 months (longest 261 months), LKB1 did not show any significant association with BCSS (Figure 3a. However, among those who received adjuvant endocrine therapy (*n* = 267), patients with ER positive disease and positive LKB1 expression (ER+/LKB1+) showed significantly better BCSS (5-year: 93%versus 74%, *p* = 0.03) (Figure 3b). However, in the absence of adjuvant endocrine therapy, LKB1 did not produce any significant impact on survival (*p* = 0.85). For patients with positive LKB1, positive expression of ER (*p* < 0.001, Figure 4a), PgR (*p* = 0.001, Figure 4b), MUC1 (*p* = 0.003, Figure 4c) and Bcl2 (*p* < 0.001, Figure 4d) was associated with significantly better BCSS but poorer BCSS in cases of positive expression of HER2 (*p* < 0.001, Figure 4e), Ki67 (*p* = 0.01, Figure 4f), CK17 (*p* = 0.02, Figure 4g) and EGFR (*p* = 0.03, Figure 4h).

### 3.5. Recurrence Free Survival

There was no significant correlation between metastases-, local recurrence- or regional recurrence-free survival and LKB1 expression (*p* > 0.05). Appendix A). Nevertheless, among patients with positive LKB1, positive expression of ER (*p* < 0.001, Figure 5a), PgR (*p* = 0.001, Figure 5b), MUC1 (*p* = 0.002, Figure 5c), Bcl2 (*p* = 0.004, Figure 5d) and negative expression of HER2 were associated with better metastases-free survival (*p* = 0.01, Figure 5e). While Ki67 (Appendix A), CK17 (Appendix A) and EGFR (Appendix A) did not show any significant influence on metastases free survival.

## 4. Discussion

Results of the analysis showed that cytosolic expression of LKB1 was associated with high-grade tumours and positive expression of HER2, Ki67 and VEGF. Within the population with LKB1 positive tumours, BCSS was significantly better in patients who received adjuvant endocrine therapy. 

The expression of LKB1 in tumours cells was previously reported as 71%, which is slightly lower than the expression reported in our population (78%) [24]. This could be explained by the age of the population, as our study included patients 70 years or older, while the previously reported data included four groups with median ages of 48, 54, 61 and 62 years [24]. In our study, the association between LKB1 positivity and the factors mentioned above suggests its role as a possible poor prognostic indicator. The association of LKB1 expression with high histological grade suggests a poor prognostic indication as reported previously, where IHC was used for the detection of LKB1 protein expression [24]. While in vitro studies reported a negative correlation between both LKB1 mutation and grade and VEGF and a positive correlation with hormone receptor expression [1,3,12,25], these studies analysed LKB1 using Western blot and RT-PCR [1,3,11]. A recently presented study suggested that the nuclear location of LKB1 was associated with better survival while cytosolic expression is an indicator of poor prognosis [25]. Another study analysed both nuclear and cytoplasmic expression (*n* = 32) on IHC in early operable primary breast cancer with ER and PgR positive expression [26]. High cytoplasmic expression was seen in 62.5% of cases, which was not associated with age, ER, PgR and HER2, but was shown to be significantly associated with smaller tumour size [26]. In this study, the nuclear expression was also associated with smaller tumours and better survival. A high LKB1 gene expression has been reported to be linked to a high ER expression [24]. The expression of LKB1 has shown its association with CD44, which is a marker of stemness in breast cancer stem cells. Previously, the available literature showed LKB1 as a factor for maintenance of stem cells in haemopoetic cells in a mouse model study, where animals with LKB1 deletion had progressive pancytopenia and reduction in haemopoetic cell proliferation [27,28,29]. In this study, LKB1 has also shown its significant association with VEGF, probably linked to tumour growth and invasion. A recent study has demonstrated the influence of LKB1 on tumour invasion and metastases in colorectal cancer and suggested it to be a potential therapeutic marker [30].

The expression of LKB1 among molecular classes or subtypes has not been reported in the literature. Though it did not show any remarkable findings in our data analyses, its relatively high expression in the HER2 over-expressing class may be linked to some relationship with growth factor pathways. As it was previously reported, it may be a potential predictor of overall survival among HER2 positive breast cancer patients [24].

The prognostic significance of LKB1 was clinically demonstrated in the group of patients who received adjuvant endocrine therapy. This is an interesting finding, which is in line with the data previously reported, which suggested that within the stress environment, 17β oestradiol stimulatesthe AMPK pathway. The same study has also suggested that LKB1 is required for the ER mediated activation of AMPK [31,32]. The tumour analysis of the patients included in the TAMRAD randomized GINECO trial (the trial compared tamoxifen versus tamoxifen plus everolimus in ER+ metastatic breast cancers). It reported that the patients with a low cytoplasmic expression of LKB1 demonstrated a 67% reduction in the risk of disease progression with everolimus (a mammalian target of rapamycin (mTOR) inhibitor) plus tamoxifen, when compared to the tamoxifen only arm [33]. On the other hand, high cytoplasmic expression was associated with less benefit with everolimus therapy, consistent with our results, suggesting the predictive role of LKB1 in patients with ER positive tumours.

In addition, LKB1 has been previously reported to inhibit the aromatase enzyme via the AMP-activated protein kinase, thus supporting its potential role in the prediction of the response to primary endocrine therapy in older women [7,10,26].

The study has limitation of not including the comparison with the younger women and not studying the gene expression and mutations in older women. Precisely defining an expression cut-off based on IHC remains a challenge as this could vary between laboratories.

## 5. Conclusions 

The available literature reported on LKB1 are primarily based on in vitro and in vivo studies using Western blot or PCR techniques. Our work describes a novel study which analysed the relationship between LKB1 and other biomarkers and clinical outcome in a large series of older patients with breast cancer by using IHC. Given the availability of IHC in clinical practice, this study possesses translational significance. The study has demonstrated significant correlations between LKB1 and biological markers of established significance. It has also shown its significant influence on the clinical outcome (notably BCSS) in older women receiving adjuvant endocrine therapy. Further studies are required to investigate and compare the expression of LKB1 in and with the younger patients and to delineate the precise role of LKB1 as a potential therapeutic target in breast cancer management. Studies are also required to understand the detailed genetics of LKB1. 

## Figures and Tables

**Figure 1 cancers-11-00149-f001:**
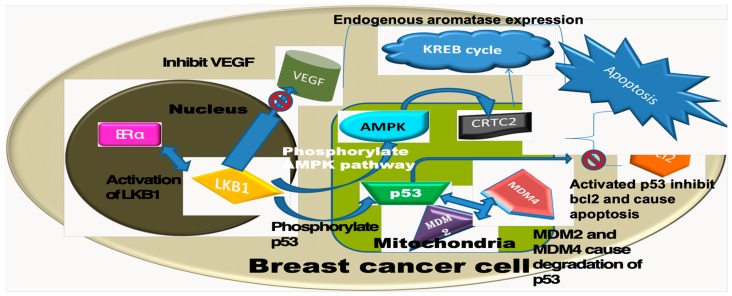
Proposed intracellular mechanism of action of LKB1 as suggested in the cell line studies [7,8,9,12,16]. VEGF: Vascular Endothelial Growth factor, KREB: Citric acid cycle, ER: Oestrogen receptor, CRTC2: Creb-regulated transcription co-activator 2, MDM2/4: Mouse double minute 2/4.

**Figure 2 cancers-11-00149-f002:**
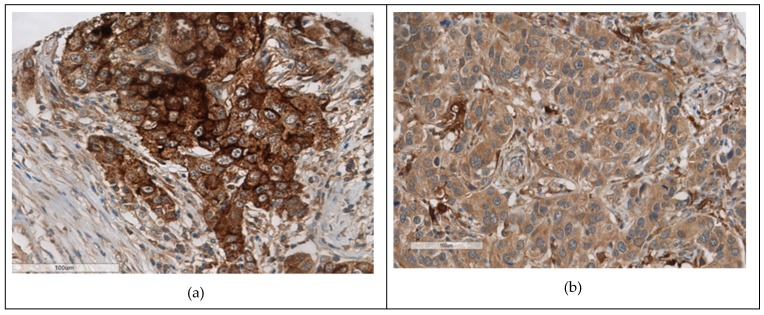
Immunohistochemichemical expression of LKB1 in Breast cancer cells. (**a**–**c**) Positive cytoplasmic expression, (**d**) LKB1 negative.

**Figure 3 cancers-11-00149-f003:**
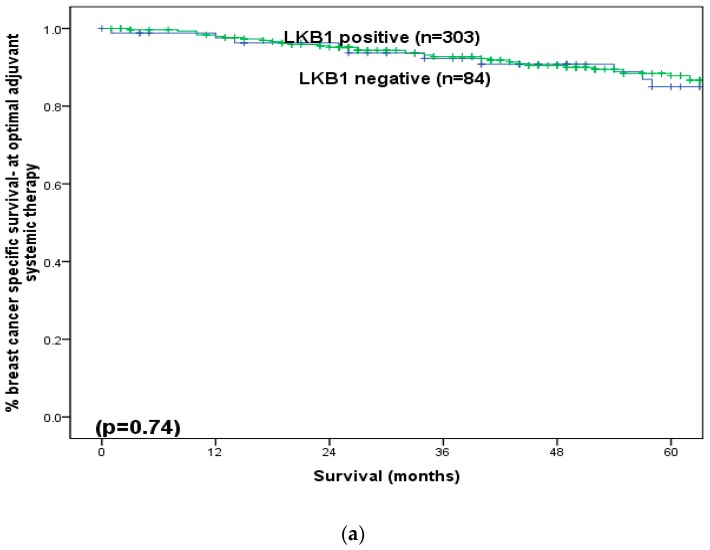
(**a**) Breast cancer specific survival according to the expression of LKB1 in older women with early operable primary breast cancer (all patients). (**b**) Breast cancer specific survival according to the expression of LKB1 in older women with early operable primary breast cancer who received adjuvant endocrine therapy.

**Figure 4 cancers-11-00149-f004:**
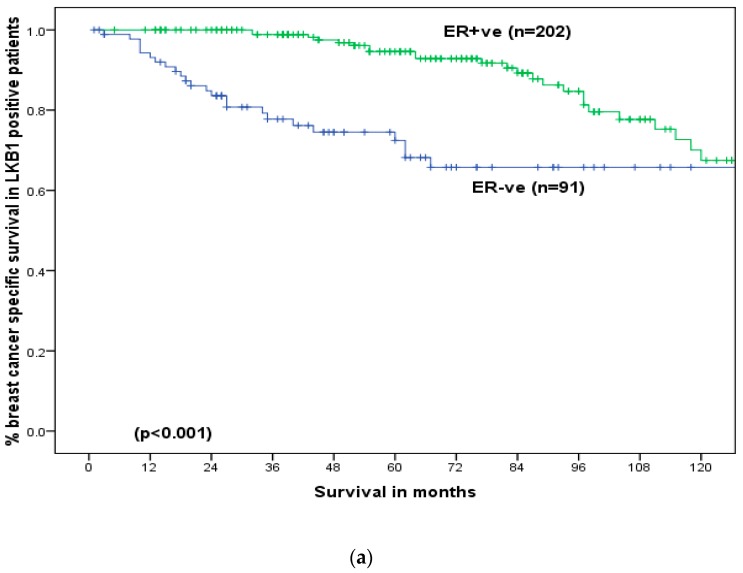
Breast cancer specific survival in LKB1 positive early operable primary breast cancer in older women: (**a**) Oestrogen receptor positive versus oestrogen receptor negative. (**b**) Progesterone receptor(PgR) positive versus Progesterone Receptor negative, (**c**) Mucin (MUC)-1 positive versus Mucin 1 negative, (**d**) B-Cell Lymphoma-2 (Bcl2) positive versus B-Cell Lymphoma 2 negative, (**e**) Human Epidermal Growth Factor Receptor-2(HER2) positive versus Human Epidermal Growth Factor Receptor-2 negative, (**f**) Ki67 positive versus Ki67 negative, (**g**) Cytokeratin 17(CK17) positive versus Cytokeratin 17 negative, (**h**) Epidermal Growth Factor Receptor (EGFR) positive versus Epidermal Growth Factor Receptor negative.

**Figure 5 cancers-11-00149-f005:**
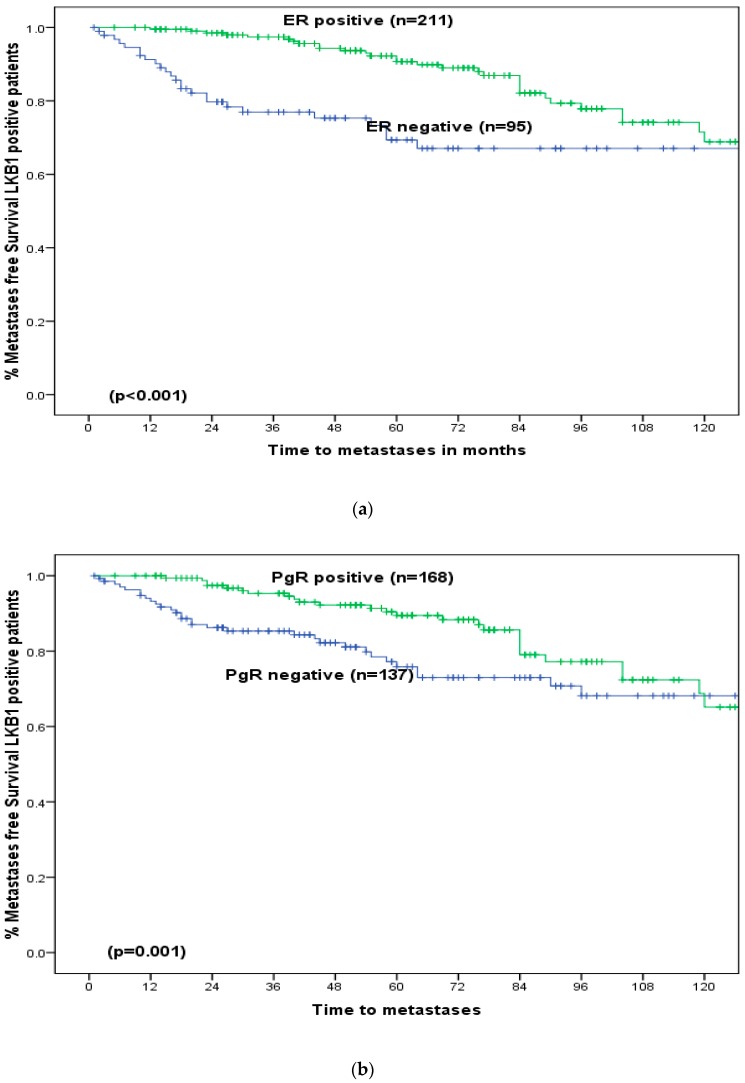
(**a**) Metastasis-free survival in LKB1 positive early operable primary breast cancer in older women: ER positive versus ER negative. (**b**) Metastasis-free survival in LKB1 positive early operable primary breast cancer in older women: PgR positive versus PgR negative. (**c**) Metastasis-free survival in LKB1 positive early operable primary breast cancer in older women: MUC1 positive versus MUC1 negative. (**d**) Metastasis-free survival in LKB1 positive early operable primary breast cancer in older women: Bcl2 positive versus Bcl2 negative. (**e**) Metastasis-free survival in LKB1 positive early operable primary breast cancer in older women: HER2 positive versus HER2 negative.

**Table 1 cancers-11-00149-t001:** Intracellular relationships between LKB1 with biological markers in older women with early operable primary breast cancer.

Biomarker	LKB1 Positive N (%)	LKB1 Negative N (%)	*p*-Value
ER + ve	211 (69.0)	61 (72.6)	0.30
PgR + ve	168 (55.1)	52 (61.9)	0.16
HER2 + ve	30 (9.6)	2 (2.4)	0.01
Ki67 + ve	114 (35.8)	20 (22.5)	0.01
MUC1 + ve	274 (89.0)	73 (86.9)	0.36
Bcl2 + ve	250 (83.1)	69 (83.1)	0.56
P53 + ve	123 (41.1)	31 (39.7)	0.46
CK5 + ve	104 (33.7)	21 (24.7)	0.07
CK5/6 + ve	128 (45.2)	33 (42.3)	0.37
CK7/8 + ve	301 (97.4)	84 (98.8)	0.38
CK14 + ve	73 (25.8)	15 (18.5)	0.11
CK17 + ve	70 (23.4)	14 (16.7)	0.11
CK18 + ve	286 (97.3)	81 (96.4)	0.45
CK19 + ve	291 (95.4)	79 (96.3)	0.49
EGFR + ve	65 (23.0)	12 (14.8)	0.07
BRCA2 + ve	160 (56.7)	35 (42.7)	0.01
VEGF + ve	234 (88.6)	56 (73.7)	0.002
CD44 + ve	61 (19.7)	25 (29.8)	0.03
MDM2 + ve	296 (96.7)	60 (74.1)	<0.001
E-Cadherin + ve	191 (63.2)	48 (57.1)	0.18
Expression of LKB1 in molecular classes of breast cancer			
Luminal A	79 (76.0)	25 (24.0)	0.09
Luminal B	56 (75.7)	18 (24.3)	
Low ER Luminal	36 (81.8)	8 (18.2)	
All low expression/normal like	13 (76.5)	4 (23.5)	
Basal Like	19 (73.1)	7 (26.9)	
HER2 positive	25 (96.2)	1 (3.8)	

ER: Oestrogen receptor, HER2: Human Epidermal Receptor 2.

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
