# Peer review of "Liver Kinase B1—A Potential Therapeutic Target in Hormone-Sensitive Breast Cancer in Older Women"

_cancers, 2019, doi:10.3390/cancers11020149_

Round 1

Reviewer 1 Report

In this study, BM Syed et al. analyzed the prognostic value of liver kinase B1 (LKB1) in 407 women (at least 70 years old) with primary breast cancer using immunohistochemistry (IHC). They showed that the level of LKB1 is not significantly correlated with the breast cancer specific survival (BCSS) of all patients but positively associates with the BCSS of the patients who received adjuvant endocrine therapy. Overall, the study was well designed and performed. But before the manuscript can be recommended for publication, the authors need to address several issues.

1 Representative IHC staining images of LKB1 positive and LKB1 negative should be shown.

2 Is the metric to classify positive/negative expression of other biomarkers such as HER2, Ki67 and VEGF consistent with LKB1 classification? Please clarify.

3 What is the correlation of LKB1 level with BCSS in patients who didn't receive adjuvant endocrine therapy? Addressing this question might help understand why there is no significant correlation of LKB1 with patient BCSS in all patients.

4 The prognostic value of LKB1 may vary among different subtypes of breast cancer (Lumina A, B, Her2+, triple negative etc.) as suggested by reference 24. Is there any similar result observed in this study?

4 There are various types of survival analyses that could be performed to evaluate the prognostic value of biomarkers. Could the authors analyze the association of LKB1 expression with patient metastasis-free survival or relapse-free survival? 

5 Could the authors comment on the potential role of LKB1 in modulating epithelial-mesenchymal transition (EMT) and stemness? EMT and stemness can significantly promote metastasis and relapse. The authors already showed the correlation of LKB1 expression with stemness markers such as CD44 and EMT markers such as E-cadherin.

6 On page 4, the sentence “LKB1 positive expression was associated with significantly better BCSS in patients with positive expression of ER (p<0.001, Figure 2a)” is confusing. Fig. 2a shows that high expression of ER significantly associates with better BCSS in LKB positive patients. Please rephrase the interpretation of figure 2.  

8 On page 6, the authors mentioned that “High cytoplasmic expressionwas seen in 62.5% cases, which was significantly associated with smaller tumour size [26]”. However, it is showed that Nuclear LKB1 expression was a marker of good prognosis. It was associated with smaller tumors”in reference 26.  Please correct the typo.

9 On page 7, the author mentioned that “In this study the nuclear expression was associated with better survival. The low LKB1 gene expression has been reported to be linked with high ER and PgRexpression [24]”. Please confirm that whether LKB1 gene expression has been shown to be associated with high PgR expression in reference 24.  

6 The potential limitations of the current study need to be discussed. For example, the cutoffs to define LKB1 positive samples may affect the results.

Author Response

Thanks very much for constructive feedback.

The comments have been addressed and the revised manuscript is attached here. 

Reviewer 2 Report

In this manuscript, Syed et. al., demonstrate the expression of Liver kinase B1 in older women suffering from breast carcinoma. They find in these patients, a higher expression of LKB1 correlates with high EGFR, VEGF, Her3 etc. The findings merely illustrate the expression characteristics of LKB1 in these patients without a clear cut demonstration of the prognosis of high LKB1. It has been reported that patients with higher LKB1 expression respond better to advanced hormone therapy. The rationale and the scientific basis for this has not been again demonstrated. As such, the manuscript is of clinical importance, however, the lack of mechanistic insight evidence showing the importance of LKB1 in breast cancer associated older women is a major weakness of this study. Specific points: 1) Have the authors compared younger women with BC versus older women? Does LKB1 expression vary with age? If so why? 2) Where is Lkb1 localized in BC tissue sections? The authors mention about nuclear LKB1 asosciated with prognosis, however, these expression data must be shown. 3) At this stage, the correlation of LKB1 with other markers are only based on expression data. What does these really mean in the context of cancer? For instance, what is the basis of Lkb1 and EGFR co-expression? 4) There are a bunch of studies that have analyzed LKB1 expression in BC. It is unclear to me as to what specifically is the major achievement of this current analysis and how does it add to what we know about LKB1 in breast cancer.

Author Response

Thanks very much for valuable comments.

The revised manuscript is attached here. 

Round 2

Reviewer 1 Report

The authors only partially addressed my concerns. Before the manuscript can be recommended for publication, the authors need to address the following issues. Please kindly provide a clear and well-written response letter with the point-to-point basis details of the changes made in the revised version according to the referees’ comments.

1.    Regarding the comment “Representative IHC staining images of LKB1 positive and LKB1 negative should be shown”, where are the images A, B and D? I didn’t see where these new images were referred to in the main text.

2.    Regarding the comment “There are various types of survival analyses that could be performed to evaluate the prognostic value of biomarkers. Could the authors comment on the association of LKB1 with patient metastasis-free survival/or relapse-free survival?”, where are the metastasis-free survival/or relapse-free survival curves? All survival curves shown in the current manuscript are breast cancer specific survival results.

3.    Regarding the comment “High cytoplasmic expression was seen in 62.5% cases, which was significantly associated with smaller tumour size [29]”. However, it is showed that “Nuclear LKB1 expression was a marker of good prognosis. It was associated with smaller tumors” in reference 29.  Please double check the reference here and correct the statement accordingly.

4.    Regarding the comment “In this study the nuclear expression was associated with better survival. The low LKB1 gene expression has been reported to be linked with high ER and PgR expression [27]”. Did the reference 27 here talk about that LKB1 gene expression associates with high PgR expression?

Author Response

Comment
Response

Regarding the comment “Representative IHC staining images of LKB1 positive and LKB1 negative should be shown”, where are the images A, B and D? I didn’t see where these new images were referred to in the main text.

Methods under scoring section.

Images were not visible due to track changes now revised

Regarding the comment “There are various types of survival analyses that could be performed to evaluate the prognostic value of biomarkers. Could the authors comment on the association of LKB1 with patient metastasis-free survival/or relapse-free survival?”, where are the metastasis-free survival/or relapse-free survival curves? All survival curves shown in the current manuscript are breast cancer specific survival results.

Done- Under relapse free survival and Figure 3 and 4

Regarding the comment “High cytoplasmic expression was seen in 62.5% cases, which was significantly associated with smaller tumour size [29]”. However, it is showed that “Nuclear LKB1 expression was a marker of good prognosis. It was associated with smaller tumors” in reference 29.  Please double check the reference here and correct the statement accordingly.

Revised in discussion

Regarding the comment “In this study the nuclear expression was associated with better survival. The low LKB1 gene expression has been reported to be linked with high ER and PgR expression [27]”. Did the reference 27 here talk about that LKB1 gene expression associates with high PgR expression?

Revised in discussion

Reviewer 2 Report

The authors have made some revisions according to the prior suggestions. and hence I am now recommending this manuscript for publication

Author Response

Thanks for your kind consideration. 

Best regards

Round 3

Reviewer 1 Report

The authors have appropriately addressed my concerns.